# A Qualitative Exploration of Parents’ Perceptions of Risk in Youth Contact Rugby

**DOI:** 10.3390/bs12120510

**Published:** 2022-12-14

**Authors:** Eric Anderson, Adam White, Jack Hardwicke

**Affiliations:** 1School of Sport, Health and Community, University of Winchester, Winchester SO22 4NR, UK; 2Department of Sport, Health Sciences and Social Work, Oxford Brookes University, Oxford OX3 0BP, UK; 3Centre for Physical Activity and Life Sciences, University of Northampton, Northampton NN1 5PH, UK

**Keywords:** concussion, rugby, cognition, parental attitudes, risk perception

## Abstract

The purpose of this study was to explore the understandings and perceptions of risk related to brain trauma amongst parents of children that play contact rugby. A qualitative approach was taken, using semi-structured interviews with 7 mothers and 27 fathers of children that participate in contact rugby. A thematic analysis of data suggests that parents used two primary cognitive strategies to process the risk they consented to with their children’s participation in rugby; (1) minimalizing rugby risk to be equivalent to less injurious sports; and (2) elevating physical and social advantages above what they think other sports are capable of providing. From the findings it is suggested that parents who permit their children to play contact rugby are both aware of the high risks of injury in the sport, but simultaneously utilize two cognitive distortion techniques to rectify the dissonance caused between their choice to have their children play, and the salient number of concussions they observe. These results suggest that it will take properly informed consent, inclusive of concussion rates compared to other sports, in order to reduce cognitive distortion and effectively communicate risks associated with participation in contact rugby.

## 1. Introduction

Defining an acceptable level of sporting risk is a complex issue, and there is no universal definition of an ‘acceptable risk’ [1]. Cognitive perceptions of risk are complicated by the variance in whether a risk occurs through compulsion or whether it is voluntarily engaged with. Philley [2] shows that people tend to accept higher voluntary (self-imposed and self-controlled) risk levels than if the same risk level were imposed involuntarily (imposed without consent or prior knowledge, such as natural disasters), and Starr [3] and Fuller [4] show this figure to be up to 1000 times greater. 

Perceptions of risk are also important to understand, particularly as they are malleable and change over time and culture in line with different worldviews [5,6]. Risk perception refers to ‘people’s beliefs, attitudes, judgements, and feelings, as well as the wider social or cultural values and dispositions that people adopt, towards hazards and their benefits’ [7]. Importantly, it concerns the subjective assessment of risk and the potential outcomes, positive and negative, of engaging in a given activity [8]. Adams [9] discusses how, from an early age, people develop cognitions to process and perceive risk in everyday activities, such as walking or crossing the road. However, perceptions of risk can often diverge from objective risk indicators because of cultural norms, personal biases and perceived benefits [9,10]. 

When risk involves children, however, matters become more complex as it raises questions of consent [11]. Typically, children are eager to engage in sport and recreation, with this willingness formed long before they are mature enough to make fully educated and autonomous decisions [12]. Neuromaturation research shows that the frontal lobes, which are responsible for risk perception and analysis, are not fully developed until a person is in their twenties [13]. Because children are too young, however, to cognitively evaluate the potential negative and long-term consequences associated with something they enjoy participating in, parents make decisions for them [14]. In individualistic cultures, such as post-industrial Western nations, issues of risk and the responsibility for the care and safety of children is increasingly placed on parents [15]. This raises questions about whether parents will subject their children to greater risk when it comes to their children ostensibly consenting to play dangerous sports. 

This also raises a question of a parents’ ability to provide informed consent for their children to participate in sports that carry a high risk of injury. In the general sporting context, consent is implied through participation, and thus any associated risks of the sport are accepted [11]. Weinberg [16] states that for consent to be informed, then clear and transparent information on the risks and benefits of the activity should be freely available to participants and those making decisions on participation. However, Channon and Matthews [17] suggest that cultural norms in sport and the involvement of third parties can disrupt the clear communication of information to allow for informed consent. For example, World Rugby was criticized for the presentation of misleading injury data and publicly requested to retract this information [18]. Use of erroneous data regarding injury risk would limit parents’ ability to make informed decisions on their child’s participation in contact rugby.

As Channon and Matthews [17] suggest, cultural norms in sport also complicate the understanding of levels of acceptable risk and communication of informed consent. The intersects of pain, injury, and risk in sporting cultures has been of interest to sport sociologists for over 30 years [19]. Nixon [20] presented the argument that sporting contexts operate within ‘cultures of risk’ which encourage athletes to accept risk as the norm in sport. Taking and accepting risks forms part of what Hughes and Coakley [21] term ‘the sports ethic’. This refers to the cultural expectation of athletes to sacrifice their bodies and health for sporting success. This is something that would be deemed deviant in broader society but is normative in sport [21]. 

In this environment, the presence of risk and encouragement of risky behaviours is normalised and thus athletes will often continue in competition despite pain and/or injury [20]. Findings consistent with this thesis have been reported amongst amateur rugby players, where players were found to trivialise concussive injuries and present being ‘head strong’ to adhere to the sports culture [22]. Thus, parents with children participating in many competitive sports, including contact rugby, must also navigate an environment steeped in strong cultural norms in which risk and injury is accepted in levels that deviate from broader society. 

Contact forms of rugby football serve as an extreme, yet popular, example of sporting-risk for children. Rugby (union and league) exists as the most commonly played contact sport in England. Amateur adult Rugby Union at the community level is argued to have a “high risk” of injury compared to other sports [23]. Whilst limited epidemiological data exists in the youth context and community youth settings, where data is available, a comparable risk of concussion between the adult community game and the youth game is shown [24]. 

Research from the community, academy, and independent school settings on youth Rugby Union [25,26] has found most time-loss injuries occur as a result of the tackle. Further, an epidemiological study of school teams found 37% of 825 children suffered a time-loss injury over the course of a season [27]. Almost half (49%) of those injured required 28 or more days away from play and 19% of injuries (*n* = 81) were concussions [27]. Further evidence of the injurious nature of the sport comes from Kirkwood et al. [28] who reported the three main sports that resulted in hospital admissions for males under the age of 19 years were football (soccer), Rugby Union, and Rugby League. 

In a systematic review of twenty-three studies of concussion in sport, Pfister et al., [29] found, per 1000 athlete exposures, “The three sports with the highest incidence rates were rugby, hockey and American football at 4.18, 1.20 and 0.53, respectively”. In another systematic review of twenty-five studies conducted by Kirkwood, Parekh, Ofori-Asenso and Pollock [30] on concussion, in both rugby league and rugby union, players under the age of twenty-one had an incidence range of 0.2 to 6.9 concussions per 1000 playing-hours. Kirkwood et al. [30], expand on this, detailing the likelihood of becoming concussed per season to be 7.7%, stating, “There is a significant risk of concussion in children and adolescents playing rugby union and rugby league.”. Freitag et al. [31] also suggested that the probability of youth rugby players getting injured over a season was between 9% for U9–U12 age groups and 98% for the U18 age groups. 

Furthermore, a recent scoping review concluded that contact rugby union presents high rates of injury and concussions relative to other sports [32]. The researchers also conclude that the evidence is either ‘mixed or unclear’ on contact forms of rugby offering any unique physical and mental health well-being benefits, over non-contact forms. This begs the question of why contact is required in the children’s sport, particularly when considered in school physical education settings [11,33]. Understanding perceived risk in contact rugby is also complicated by the fact that much of the damage that is caused to the brain on impact is not seen [34]. Unlike musculo-skeletal damage, which is normally immediately visible, concussions can occur and yet go undetected. 

These concussive sporting injuries are but one of two major concerns for children’s brains in sport, alongside long-term degenerative issues. In 2019, Mackay et al. [35] showed that ex-football players in the UK were at a significantly elevated risk for cognitive diseases that cause dementia, compared to a population control group. Accordingly, in 2020 the English Football Association banned heading the ball in practice, and strongly discouraged its use in games, for children under 12 [36]. A similar preventative policy has not yet been brought in by the governing body that promotes rugby. The issue of brain trauma in contact rugby becomes even more significant with recent research highlighting a causal relationship between repetitive head impacts and the neuro-degenerative disease, Chronic Traumatic Encephalopathy [37]. 

Debates thus exist as to the acceptable levels of risk towards children in rugby between “… those charged with administering and promoting rugby union and those wanting tackles and other forms of harmful contact to be banned” [38]. Opinions on the matter are thus informed by a combination of scientific inquiry, social lore, and cultural norms. Fuller [39], attempts to mitigate some of these problems with the dictate that an acceptable level of risk to children should be no greater than the levels of risk encountered daily. Debates about risk are also complicated by those arguing for more risk in children’s lives. Russell [40] expands on this, believing physical risk to contribute an important value to a child’s wellbeing, a belief stemmed from two points. Firstly, experiences of risks aid a healthy development through preparation for future physical endeavours and secondly, how a fundamental part of childhood is testing one’s physical capabilities, aiding the child to discover ‘who they are’ [41]. 

With opinions on the topic suggesting that children ought to be exposed to high levels of risk, to opinions that they should be exposed to no elevated risk beyond daily life [39], parents navigate a complex and rapidly changing sport when they consent to their children playing what researchers agree to be the most dangerous team sport commonly played among British children [42,43,44,45,46]. Therefore, the objective of this study is to explore the perceptions of risk among parents who enrol their children in contact rugby, and how they understand and manage this risk. 

## 2. Methods

### 2.1. Study Design 

This research study used qualitative methodology due to the exploratory characteristic of the research design [47]. The study used a qualitative approach to allow for an in-depth and more detailed understanding of parents’ perceptions of risk than would have been possible via a quantitative approach. Furthermore, the research sought to understand the lived experiences of the participants and to gain rich, contextual, data which highlights the approach taken as the most viable to allow a more comprehensive understanding of the phenomenon being studied. Specifically, an inductive approach was taken, and semi-structured interviews were decided to be the most viable qualitative method for understanding the emotional terrain concerning risk perception in this setting. The rationale behind the use of semi-structured interviews concerns flexibility in inductively approaching exploratory research, and opportunity to explore answers and ask follow-up questions [48]. It was hoped that this method would provide more meaningful answers compared to what might have occurred with surveys or structured interviews [49]. This research followed standards for reporting qualitative research (SRQR) guidelines which can be seen in Appendix A. 

### 2.2. Participants 

The study employed a purposeful sampling procedure [50] in order to explore parental perceptions, understandings and management strategies of risk related to their child’s participation in contact rugby. A local rugby club in the South-East of England was identified and the researchers received gate-keeper control to interview parents of active youth players. Whilst a purposeful sampling procedure was used to identify the target sample, opportunity sampling was used when in this setting to recruit participants. The researchers attended male rugby matches to interview a total of thirty-four parents of forty-eight children, whom were all aged 13–16 yrs, currently participating in two rugby teams. The variance between the two figures concerns the fact that some parents had more than one child playing rugby. These children were not, because of age differences, playing on the same team, but all were playing contact rugby. 

The sample for this study was limited to parents who attended the rugby matches over the three weekends data was collected. As a result of this, there is not homogeneity of gender of participants, with twenty-seven males and seven females. This imbalance is likely a reflection of the fact that fathers were more likely to attend their son’s rugby matches. Despite the gendered variance in who attends matches, mothers were more eager to be interviewed. Of the six parents who approached the research team for interview (after hearing about it from friends) five were female. It is therefore suggested that the data represented is more generalizable to males than females. Furthermore, the club was located in an affluent urban area with the socio-economic backgrounds of the families being broadly middle-class. Whilst a gendered or socio-economic analysis in this research is not provided, it is suggested this would be rife for future research. 

### 2.3. Procedures

The semi-structured interview schedule consisted of open and closed questions across five sections: demographics, participation in rugby, injuries, concussion and awareness. Questions were carefully designed to help to assess parent’s understanding of and acceptance towards injury and particularly head trauma. Researchers were permitted to take each question into an open-ended discussion, however. This helps explain the variability of interviews. The shortest interview was just 20 min, and the longest, 60. Most interviews lasted about 35 min. Data was collected using a recording device from parents of players and transcribed verbatim to Microsoft Word at a later date. Interviews were conducted in the club’s social spaces, while the parents interviewed attended their children’s matches. The interviews were undertaken by the first and second author. 

### 2.4. Data Analysis 

A thematic analysis was used to analyse the data, with the six-step process outlined by Braun and Clarke followed [51]. Data familiarization, which occurred through analysis of interview transcripts and hand-written notes that accompanied the interview schedule, was conducted first. From this, the first wave of coding was co-created between the first and second authors. Theme development occurred in relation to a thematic reading of the interviews, by the same two researchers. This process involved categorizing recurrent data into themes [52] that underpin the research aims [53]. After thematic coding was agreed upon, data was subjected to axial coding, a qualitative research technique that involves relating data together in order to construct codes, categories, and subcategories grounded in participants’ voices within the data. Interview content and themes were then independently assessed and reviewed by the third author to assist and finalise the thematic analysis process. Following this, final themes were agreed on between the research team to present the data. 

### 2.5. Researcher Backgrounds

All authors are experienced in qualitative research, data collection and data analysis. Furthermore, the authors are experienced specifically in undertaking interview-based research on sensitive topics which brings a strength to this research. AJW has extensive previous experience as part of a professional rugby organization and has held a number of coaching and rereferring roles in youth rugby. This experience aided the research teams’ access and acceptance into the rugby club from which participants were recruited. AJW’s experience in rugby helped to build rapport with the participants. EA and JH do not have a background in rugby and thus acted as critical colleagues to encourage reflexivity in the interpretation of the data. The researchers had no relationship to the rugby club or participants prior to the research. All researchers are male, and although a gendered analysis is not undertaken in this study, this researcher characteristic may have influenced participants responses to questioning as well as willingness to participate in the research. 

### 2.6. Ethics

Institutional ethical approval was obtained, as well as consent of the gatekeepers of the rugby teams (coaches) concerned. All relevant ethical procedures of the British Psychological Society were adhered to. This includes the fact that all participants were provided an information sheet detailing and explaining the study before written informed consent was obtained. Strongly emphasized, was how all collated data would be kept confidential and anonymous, with the right to withdraw at any time without consequence also highlighted. All data has been anonymized; numbers are used to identify participants instead of pseudonyms. There were no financial or other incentives to participate in the interviews. 

## 3. Results and Discussion 

### 3.1. Evidencing Awareness of Concussive Injuries in Rugby

Parents interviewed demonstrated they were cognitively aware that concussion was frequent in their child’s sport. Evidencing this, every parent interviewed indicated that they had frequently witnessed concussions among players. When asked ‘Have you witnessed any concussions on the rugby pitch?’, P22 said, “I’ve seen lots of head injuries” whilst P26 recognised it, “Dozens of times”. Similarly, P4, P10 and P13 said, “We’ve had a few boys with concussion”. “There were a couple last week, where a few boys got knocked out”, and “I know quite a few children who’ve had concussion”, respectively. Furthermore, P20 refers to one game where, “It was a friendly, but the other side were very aggressive in their approach… I know in that game there were three concussions”. P23 explained how at a tournament his son was playing in, they “… saw a boy get a bit of a knock… He did go off to hospital… he looked a bit injured, a bit bashed”. 

When it came to injury and concussion among their own children, results showed that 84% had sustained an injury (informing perceptions of risk in general) and 31.3% of participants sustained either a medically examined or a suspected concussion, the most frequent reported injury accounting for 38.4% of injuries reported in this study. These findings fit with epidemiological research on injury rates in youth rugby [27]. 

Given that nearly a third of the parents interviewed reported their child having at least once been concussed in rugby, it was expected, and found that parents interviewed had experience of their own children’s concussions. P23 said:

My 16 and 13-year-old both recently had concussion. My eldest was in a tournament, came off with concussion, they were short of players and put him back on the pitch. He then got a neck injury and was later in A & E with his head taped, you know, an X-ray and scan.

Similarly, P15 said:

My youngest had to take a while off through concussion. He tried to tackle someone and as he went in for the tackle, they turned so their knee came up into the head.

Three parents; P14, P20 and P22, also explained how their child had to miss school as a consequence of concussion. P14 explained:

Effectively two people tackled him, and his head hit on someone else’s head. He had a couple of days off school, he wasn’t really right but within a couple of weeks he was fine again.

### 3.2. Adjudicating Risk 

Findings suggest that parents use cognitive distortion strategies in order to mitigate fear of brain trauma as a result of playing rugby. Cognitive distortion refers to habitual ways of thinking that are often inaccurate and negatively biased, and can influence decision-making [54]. The most salient analysis derived from the coding concerns what is described as cognitive equalizing. For example, P10, P12, P15, P29 and P34, respectively, use cognitive minimizing and equalizing to mitigate the risk of injury caused through rugby participation by comparing it to the risk of sports participation as a whole. “There’s a risk element with any sport”, “Yes it’s a risk, but there’s a risk to a lot of sports”, “They get injured, but you can get injured playing any sport”, “I think there’s risks inherent in any sport or doing any sort of physical activity” and “With all the sports, there is certain degrees of risk I guess, you just have to manage the risk”. Hence, these parents flatten actual rates of injury risk to fit their cognitive idea of equality of risk across sports. 

It cannot be determined from the interviews if this is, however, a purely cognitive distortion strategy. They may actually believe that all sports do offer the same level of risk. It can be said, however, that given these two propositions, it is the first which is empirically validated. As the literature review highlighted, and Pfister and colleagues [29] confirm with a systematic review of twelve different sports, rugby has one of the highest incidence rates of concussion across popular sports played by British children. Again, rugby is found to have 4.18 per 1000 athlete exposures, hockey, and American football at 1.2 and 0.53 per 1000 exposures, respectively. 

This finding is important for understanding how to educate parents about risks associated with contact rugby. If parental consent is diminished simply through education of the elevated statistical possibility of head trauma—and other muscular-skeletal injuries—then an education campaign alone may delimit the percentage of British parents willing to let their children play the sport. 

However, interview data informs us that the minimization of head injury risk is more likely a cognitive strategy for dealing with data/information that challenges parents’ risk-permitting schema through participation in a dangerous collision team sport. For example, P1 mitigated the risks to their children by saying, “It is a bit of a worry, but probably no more than cycling on the road” and participant 3 said, “It’s part of the game (the risk of injury), like I said earlier, as many injuries that you can get from riding a bike”. Yet, comprehensive data from New Zealand [55] found cycling to have only 100 injuries requiring hospitalization per 1 million exposures. Conversely, rugby had 52,875 game injuries per 1 million exposures. This finding is likely due to the perceived benefits parents deem their children gain from participant, which is discussed further below. 

With no policy-directed definitional guidance towards an acceptable sporting level of risk or injury, research shows acceptance levels to be contextually subjective [46], based upon values, abilities, personality, environment, and experiences [38]. Accordingly, some of the parents in the current study utilized a second strategy of simply minimizing the risk. P12 minimized risk, by saying, “I suppose there’s a little bit of danger of getting injured… but it’s unlikely to be life threatening”. P1 said, “I don’t think a knock to the head is anything to worry about much”. Participant 15, and four other fathers, agreed, “I got concussion playing rugby, and I’m fine”.

When asked, “What would cause you to take your son out of rugby”, P12 replied, “A feeling that the coaches didn’t have the best interests at heart, not the right team ethic as well”. A few parents seemed aware of elevated risks to the game, but they nonetheless adjudicated that the risks were worth whatever it is they thought the sport uniquely offered their son(s). Concerning risk of injury, P32, for example, used both equalizing and minimizing strategies:

It’s a part of the game, they are going to get injured eventually, it’s highly likely. There is always that risk with any sort of sport you can get injured. Rugby’s a physical sport, but I wouldn’t say there’s any really bad negatives to the game… It’s not really a big issue. I mean yes, it is a concern, but you do it partly because of that risk factor of the game, it’s part of the challenge of it. Part of that adrenaline thing and a fact of why you want to play it really.

P2 also explained how this risk of injury and physicality has left him and his wife ‘concerned’ but that they “… get over that, that’s not stopped us or anything” in allowing their child to play. Likewise, P8 stated how, “On balance I still think it’s worth doing” and “… probably a risk worth taking” if “… doing it in a structured environment where there are first aiders around.” 

P9 suggested, “I suppose as a parent you worry. I know there’s minor injuries, but you could potentially get quite major injuries”. Despite this recognition, he later shared that the risk of injury did not change his perception of the game. This is comparable to P11 who explained, “I’m worried about it and always have been; but I wouldn’t stop them playing because of it. I’d just like them to be more aware about it.” 

Following the recognition that his son experienced a concussion during rugby, P14 stated, “It wasn’t particularly pleasant as a parent but it’s part and parcel of the game. If you think you’re going to get that injury you’d never do anything would you?” Likewise, P27 expressed, “It’s a physical contact sport… you break bones, you bump heads. It’s part of the game. I think people get hurt whatever… life comes with no guarantees”. 

P18 said:

It’s not nice to see kids down and hurt and ambulances turn up… but I think we all know the risks. You can’t act oblivious to it; you just don’t think about it too much. It’s a contact sport, you’re never going to get away from it.

P33 said, “I know there’s serious injury, but you know the more people that play the more injuries that are going to happen, I don’t see it as a big negative no.” Agreeing with this, P26 can be quoted as saying:

He will get injured, there’s no two ways about it, he’ll pick up numerous injuries, that’s the nature of the game, but I’m not going to worry about it. It makes the game what it is and make the players who they are. If there wasn’t a risk the game would be ruined. It is a risk of the game that you could get hurt. The risk of injury is a negative but even with the long list of injuries I wouldn’t change a thing.

Lastly, P34 said that, “There are risks there, especially the way he plays cause he’s a forward. He’s always in the thick of it, but it’s not something that’s going to bother me.” 

When parents minimise risk or equate it to being on par with other sports, these influences remain uncontested with real data. Parents therefore only consent to permitting their children to play; they do not make informed consent. The lack of informed consent, combined with a history and lore of the sport, made their cognitive processing of risk easier to minimise, despite the saliency of concussion within the sport. 

To make sense of this cognitive processes, Cognitive Dissonance Theory is utilized. The theory is a proven heuristic tool for analysing the contrast between two or more incompatible cognitions—and the behavioural implications of this inconsistency [56,57,58]. The authors argue that, in the case of rugby parents, dissonance emerges from their emotional beliefs of rugby as being physically and emotionally healthy, and the often salient somatic (biological) reality of damage to the brain and body. Cognitive dissonance theory is appropriate in this context because it has frequently been used to explain how people deal with the tension caused by such variance [57,59]. 

It is suggested that, similar to other studies employing Cognitive Dissonance Theory in this fashion [60], participant’s competing and contrasting desires and knowledge produce tension that ultimately lead most participants to find catharsis through cognitive distortion of the reality of danger. Parents also rectified this dissonance through distortions, or schemas, which highlighted a variety of social, physical, and psychological benefits that rugby seemed to—uniquely—provide their sons. This, then, is a case of underplaying risk and overplaying benefit through cognitive distortions, which are thoughts that cause individuals to perceive reality inaccurately [61].

### 3.3. Beliefs about the Benefits of Rugby

Choice to permit children to play a risky sport has thus far been described as occurring through cognitive minimising or equalising rugby as possessing no more risk compared to other sports. This alone, however, may not be enough to offset the fact that, phenomenologically, concussion is salient on their child’s teams. As the results showed, roughly a third of the parents have had a child who suffered a diagnosed concussion. 

Results suggest that these socialization processes of mitigation exist alongside an over-emphasis on rugby as maintaining (1) social; (2) physical; and (3) mental health benefits that are unique to rugby. Similar findings were reported by McGlynn et al. [62] who studied motivations for parents to permit their children to play youth tackle American football. In the current study, a total of thirty-three of the thirty-four parents identified social benefits as a factor for their rugby participation decision. Of academic interest, however, is that while rates of concussion can be, and have been repeatedly, shown to be highly elevated for rugby participation, no research shows that rugby offers unique mental, social, or physical health benefits compared to other team sports [63]. Further, contact forms of rugby offer no unique well-being benefits over non-contact forms of rugby [32]. 

Parents identified rugby as an activity which allowed fostering of new friendships groups. P4 states his son has been enabled to “… meet other kids from different backgrounds, different areas... all that social part of it I think is useful.” P5 acknowledged how his son has “… formed a good bunch of friends, it’s that sense of belonging to a club which is important” and likewise, P32 recognised how rugby enabled his son to integrate with new friends.

P23 said that, “Communication skills that children learn in the sporting environment are just as important as those learnt in education, especially in their early stages as education.” P28 shared her belief that rugby has helped her son with Asperger’s to develop social skills, stating, “Before rugby he couldn’t really be with other kids, now he’s a forward so he’s in the scrum so he’s got no choice, it’s been invaluable really.” Lastly, P31 shared her belief that, “Rugby is really important for development of social skills”. 

In addition to the perception of social benefit there is a perception of elevated physical benefit. Twenty-six of the thirty-four interviewees referred to how rugby participation has provided a platform for physical development in relation to both health and physical benefits through an increase in general fitness. The participant’s lore for their sport is reflective of a culture that socially merits the playing of team sports [64]. As previously mentioned, this is common across sports. However, there is no research showing that the sport of rugby, or the act of tackling, has any unique positive impact on a child [65]. Despite this, some parents highlighted its seemingly unique importance. 

P33 suggested rugby to be, “A great game to keep fit”, but could not suggest how rugby was better than any other sport for this. P6 and P27 each argued that rugby prevented their children from leading sedentary lifestyles, stating that, “They spend most of their lives staring at computer screens at home, so it gets him out, he doesn’t turn into a couch potato” and “I think for them it’s very much being active rather than staring at an iPad or watching the television, it’s engaging in sport”, respectively. Neither could suggest how this sport might be better for that than others. 

Finally, twenty-two of the thirty-four parents referred to mental benefits that they believed their children derived from playing rugby. It is here that the majority of parents maintained that that rugby offered unique abilities. P4 expressed that, “I think the discipline around rugby in relation to the way you treat the referee and learning to deal with decisions whether they’re right or wrong is pretty unique in rugby.” Likewise, P9 said, “What’s good is the discipline, even though it’s for a very physical game, they try and control discipline as much as they can.” This participant believed that the extreme physicality of it promoted discipline more than endurance sports. P33 agreed, “Rugby is a very physical sport…it teaches them more about discipline than other sports. Finally, P14 stated, “It’s just knowing a bit of respect for your teammates, respect for the senior adults and the whole discipline around it. That is a rugby culture thing that does not exist as well in other sports”.

This focus on perceived mental benefits, in particular discipline, was also reported by McGlynn et al. [62], where parents who permitted their children to play youth tackle football cited benefits of teaching children discipline and ability to follow authority. However, the mechanism in which contact sports can uniquely provide these perceived benefits over other non-contact forms of physical activity is not clear or supported by academic evidence. Further, competitive sport encouraging discipline and obedience to authority amongst children comes with a host of potential socio-negative consequences that also must be considered [64]. 

### 3.4. Study Limitations 

This study offers insightful findings to the phenomenon studied; however, the qualitative approach limits the generalizability of the findings and further research is required to contextualise the findings reported on in this article. Furthermore, the sample was limited to parents who attended the rugby matches over the three weekends data was collected. As a result of this, participants were not balanced by gender and the results may be more relevant to males than females. With rugby being a historically male and hypermasculine terrain, this imbalance is likely a reflection of the likelihood of fathers attending their son’s rugby matches. It is, however, highlighted that a gendered analysis of differences in risk perception would be rife for future research. The socio-economic background of the participants did not form part of the analysis and, from the researcher’s subjective assessment, most participants came from a broadly middle-class background. This is also highlighted as a limitation of this research and an avenue for future research on the topic. Finally, the current study is limited to one geographical location, in England. As such, the findings are not claimed to be generalizable internationally. Again, future research in this area conducted in other countries could reveal interesting findings related to cultural differences in risk aversion and perceptions.

## 4. Conclusions

The findings from this study suggest that parents are both cognitively aware of the high rates of brain trauma in youth contact rugby, but also cognitively distort this reality. To do this, they both flatten the risks to be equivalent to other sports and other forms of physical activity; and they elevate perceived benefits of the sport compared to other sports and forms of physical activity. Hence, their cognitive distortion operates on two levels, making their choice of sport more robust. It is highlighted that cognitive minimalizing of risk operates despite academic evidence; and that cognitive exaggeration of rugby’s perceived benefits also operates without academic evidence. In summary, participation in contact rugby occurs in a complicated cognitive terrain where parents must navigate a complex environment directed by scientific inquiry, cultural lore and sporting norms when providing informed consent for their children to play one of the most dangerous team sports played among British children [43,46,66]. The findings suggest that parents minimise risk and over state benefits of contact rugby, leading to the conclusion that it will take a stronger, and sustained, education campaign before parents begin to see the research on risk and benefits of rugby in a measured light to allow for informed consent to be possible. 

## Data Availability

Due to the nature of this research, participants did not agree for their identified data to be shared publicly.

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
