# Peer review of "A Qualitative Exploration of Parents’ Perceptions of Risk in Youth Contact Rugby"

_behavsci, 2022, doi:10.3390/bs12120510_

Round 1

Reviewer 1 Report

Dear Authors,

Please find attached the review document.

Kind regards

Author Response

We would like to thank you for your time reviewing this manuscript. The comments are very thorough and constructive, and we feel they have significantly strengthened the paper- thank you. Please find our exact responses to each point below.

ABSTRACT

1) The objective of the study should be made more specific by eliminating the

numbers/data. For example: to explore the understanding and perception of parents of young people who play contact rugby of the risk of brain trauma as a result of playing contact rugby. Deleted data should be included at a later date.

Thank you for this comment, this has been addressed in the manuscript.

INTRODUCTION

2) In this section as well as in the following ones, the references included are written differently from what is stipulated by the journal. The reference number should appear in the text and in the references section, include it in a convenient way. Change all the text in this regard.

All referencing has been updated to the required style.

3) I miss that more types of injuries that can be caused by rugby are discussed, not only concussions. This would better justify the risk of practicing this sport (subject of the manuscript).

The range of injuries that occur in rugby are acknowledged by the authors, and in the manuscript. The focus on concussion concerns the current cultural environment in which concussion is the predominant talking point in the sport and poses the most risk to long-term health.  Some further justification for this focus has been added to the manuscript.

4) The objective of the research should be clearly stated at the end of the introduction section. In addition, this should coincide with the one established in the abstract. Expressions such as: "The objective of this study is: ...." can be used.

Thank you for this useful feedback, this has now been added to the introduction.

‘Therefore, the objective of this study is to explore the perceptions of risk among parents who enrol their children in contact rugby, and how they understand and manage this risk.  ‘

METHODS

5) Include a subsection dealing with the study design. This should include the information discussed in other sections (qualitative methods, descriptive study, etc.). It is important to explain the design followed in a specific section.

Thank you for this and we agree. The methods have been reworked and a study design section added.

6) The methods subsections should be numbered (2.1, 2.2...).

This has been addressed in the manuscript.

7) More detailed information on the selection of participants should be included in the "participants" section. In addition, it should be explained whether a case study, randomized selection by convenience, recruitment by accessibility, etc., was followed.

This has been updated:

‘The study employed a purposeful sampling procedure (Tongco, 2007) in order to explore parental perceptions, understandings and management strategies of risk related to their child’s participation in contact rugby. A local rugby club in the South-East of England was identified and the researchers received gate-keeper control to interview parents of active youth players. Whilst a purposeful sampling procedure was used to identify the target sample, opportunity sampling was used when in this setting to recruit participants.’

8) In the "participants" section, it would be useful to provide information on the characteristics of the club: rural/urban, socio-economic level of the families, number of members, if there is a senior professional team .....

This section has been updated to include further information on the setting.

9) Line 199 states that an institutional permit was obtained. Was there a code or registration number? If such a code exists, it would be necessary to incorporate it into the article.

This has now been included in the appropriate section at the bottom of the manuscript.

FINDINGS AND DISCUSION

10) Change the heading title to "Results and discussion".

Updated in manuscript.

11) This section should be substantially improved. Most of the information provided deals with results. The discussion part should be improved, based on the existing scientific literature. A justification or interpretation of the results should also be incorporated, i.e., why these results may have been obtained.

Thank you for this feedback. The results and discussion section has been revised throughout to integrate more discussion as well as interpretation of the results. The authors highlight that relevant literature and theory has been used throughout the section to contextualise the results. However, we are happy to engage in further discussion regarding this.

12) Section 3.3 should have a title in which the authors do not subjectively evaluate the results. An example of the new title could be: 3.3 Beliefs about the benefits of rugby.

Thank you for this constructive comment and suggestion. We agree this better reflects the content have updated the manuscript accordingly.

13) The last paragraphs of this section should include the limitations that the authors consider on the study and future lines of research related to the subject.

A section covering this has now been included.

CONCLUSIONS

14) This section must be adapted. It should not include a summary of the introduction section. They should be precise conclusions of the highlights of the article and in relation to the research objectives.

The conclusion has now been tightened up in response to these comments. We feel this has improved this section substantially- thank you.

OTRAS CONSIDERACIONES

15) References should be numbered in the order in which they are used in the body of the document. They do not follow alphabetical order. Adapt all references according to the standards of the journal.

This has now been updated in the manuscript.

16) On numerous occasions, the first-person plural is used to provide information (e.g., line 276: "we cannot..."). Adapt everything to third person or impersonal style.

This has been updated throughout the manuscript.

17) Change some references for more current ones and incorporate new ones. 17 of the 66 references have been published since 2017, the rest are earlier. Below are some publications that may be of interest to you:

The paper and literature are focused on injury in youth rugby, therefore papers focused on this population are used throughout. Unfortunately, there is limited work on this population group hence some of the relevant publications being dated. Further, the focus on concussion is justified in response to comment (3) and outlines the exclusion of specific studies on musculo-skeletal injuries. We thank you for the publications highlighted, please find our responses to each below.

Match and Training Injuries in Women’s Rugby Union: A Systematic Review of

Published Studies | SpringerLink

Not relevant to the specific population of interest to the paper.

The Epidemiology of Youth Sport-Related Shoulder Injuries: A Systematic Review

(hindawi.com)

Not relevant to the focus of the paper.

The Incidence of Injury in Amateur Male Rugby Union: A Systematic Review and MetaAnalysis | SpringerLink

Not relevant to the specific population of interest to the paper.

Systematic review of rugby injuries in children and adolescents under 21 years | British

Journal of Sports Medicine (bmj.com)

This is already included in the paper (Reference number 31)

Epidemiology of Adolescent Rugby Injuries: A Systematic Review | Journal of Athletic

Training (allenpress.com)

This is a 2011 publication, with the above publication being more recent hence its inclusion.

Hamstrings injury incidence, risk factors, and prevention in Rugby Union players: a systematic review: The Physician and Sportsmedicine: Vol 0, No 0 (tandfonline.com)

Not relevant to the focus of the paper.

Reviewer 2 Report

1. The introduction is too extensive and contains redundant content that is not related to the problem of the work. The review of the literature on the subject is disorganized and lacks a leading idea.

2. The lack of distinction between the sexes of the respondents contains a systemic error related to the different perception of danger by men and women.

3. The lack of statistical quantitative analysis makes the work an unscientific description of opinions collected from interviews and a collection of quotes.

4. The results should be separated from the discussion and accompanied by statistical analysis. A false methodological scheme was adopted.

5. Conclusions - are not conclusions, but only a partial summary of the previous chapter, going far beyond the scope of the discussed issue. Even as a summary, they are illegitimate.

Author Response

Thank you for your time reviewing this manuscript and the comments provided. Please find our responses below.

  1. The introduction is too extensive and contains redundant content that is not related to the problem of the work. The review of the literature on the subject is disorganized and lacks a leading idea.

Thank you for this comment. The literature review has been revised to add to overall clarity. However, the authors suggest it would have been useful to highlight ‘redundant content’ as this is not clear and thus cannot be acted on.

  1. The lack of distinction between the sexes of the respondents contains a systemic error related to the different perception of danger by men and women.

The focus of the paper is not on gendered difference, and this is clearly explained in the methods section and additionally in the added limitations section.

  1. The lack of statistical quantitative analysis makes the work an unscientific description of opinions collected from interviews and a collection of quotes.

The research is clearly outlined as a qualitative research study. It is unfortunate to see such a narrow view of science, and the authors would like to remind the reviewer that quantitative approaches are just one of two dominant methodologies in science (qualitative being the other). This appears to be more of an issue with ontological and epistemological differences, as opposed to a weakness of the paper. We suggest the use of ‘unscientific’ is not constructive in this context and highlights a lack of basic understanding of scientific research methodology in general.

  1. The results should be separated from the discussion and accompanied by statistical analysis. A false methodological scheme was adopted.

 This is not standard practice for qualitative research. See above comment RE statistical analysis.

  1. Conclusions - are not conclusions, but only a partial summary of the previous chapter, going far beyond the scope of the discussed issue. Even as a summary, they are illegitimate.

The conclusion section has been revised to outline the findings from the study.

Round 2

Reviewer 1 Report

Dear Authors,

First of all, I would like to thank you for taking into consideration the contributions I proposed in the previous review.

Secondly, I would like to comment on some improvements that you should take into consideration in order to improve the manuscript you have written: 

In the previous review I made the following comment: “On numerous occasions, the first-person plural is used to provide information. Adapt everything to third person or impersonal style”. I have noticed that several modifications have been made in this regard, but there are still many other modifications to be made, for example: lines 191, 185, 498, 520... These are just a few examples. You should review the whole document in this regard, not just the examples I have just listed for you.

- In lines 468-478 they have incorporated a section on study limitations. It is true that I told them to incorporate this information, but it is not necessary for it to appear in a specific section. This information should be included in the last paragraph of the previous section (results and discussion).

- In the last section of the manuscript, references continue to be included in a format different from that established in the journal "Behavioral Sciences". References should be worded appropriately.

Finally, I would like to end by congratulating you on the manuscript you have produced.

Best regards

Author Response

Thank you again for your time on this paper. We have addressed the additional comments with specific responses below. We believe these have again strengthened the paper and thank the reviewer for this.

- In the previous review I made the following comment: “On numerous occasions, the first-person plural is used to provide information. Adapt everything to third person or impersonal style”. I have noticed that several modifications have been made in this regard, but there are still many other modifications to be made, for example: lines 191, 185, 498, 520... These are just a few examples. You should review the whole document in this regard, not just the examples I have just listed for you.

Further amendments in this regard have been made throughout the whole manuscript.  

- In lines 468-478 they have incorporated a section on study limitations. It is true that I told them to incorporate this information, but it is not necessary for it to appear in a specific section. This information should be included in the last paragraph of the previous section (results and discussion).

This section has now been changed to a sub-section ‘3.4 Study Limitations’ as opposed to a specific section. Thank you for this feedback, it has made the manuscript content more logical. We feel the subheading is required to contextualise the section, rather than it flowing straight from the discussion in the preceding paragraphs.

- In the last section of the manuscript, references continue to be included in a format different from that established in the journal "Behavioral Sciences". References should be worded appropriately.

The mistake has been corrected in the conclusion section and the repetition of the same reference was not intentional. Thank you highlighting this.

References throughout have been reviewed and should now conform to the journal’s requirements. In places the named author is included as the sentence would not function without this. For example:

'This focus on perceived mental benefits, in particular discipline, was also reported by McGlynn et al. [62], where parents who permitted their children to play youth tackle football cited benefits of teaching children discipline and ability to follow authority.' 

Should the paper be accepted, we are happy to work with the editorial team on this if required.

Finally, I would like to end by congratulating you on the manuscript you have produced.

Thank you again for your constructive comments on this work and the positive praise.

Reviewer 2 Report

Fundamental objections raised in the review were not taken into account

Author Response

The manuscript has been revised to enhance clarity on the research approach taken.